# Baseline assessment of knowledge, attitude, practice, and adherence toward antimicrobials among women living in two urban municipalities in Lalitpur district, Nepal

**Nisha Jha**[1]*, **Sajala Kafle**[2], **Mili Joshi**[2], **Aakriti Pandey**[3], **Prakriti Koirala**[4], **Shital Bhandary**[3], **Pathiyil Ravi Shankar**[5]

1 Department of Clinical Pharmacology and Therapeutics, KIST Medical College, Lalitpur, Nepal,
2 Department of Pharmacology, Patan Academy of Health Sciences, Lalitpur, Nepal, 3 Department of Community Health Sciences and School of Public Health, Patan Academy of Health Sciences, Lalitpur, Nepal, 4 Transcultural Psychosocial Organization, Kathmandu, Nepal, 5 IMU University Centre for Education, IMU University, Kuala Lumpur, Malaysia

* nishajha32@gmail.com

## Abstract

### Introduction

Antimicrobial resistance (AMR) is a significant problem in developing, low- and middle-income countries like Nepal. Community engagement can be an important means to address the problem. Knowledge, attitude, practice, and adherence of women regarding antibiotics and AMR was studied.

### Methods

This baseline study was conducted in two urban municipalities of Lalitpur district as preparation for a larger intervention study (Mahalaxmi municipality will be the intervention and Godawari the comparison/control area). The study population was women belonging to the mother's groups of 45 female community health volunteers (FCHVs) from each municipality. The study was done from September 2023 to January 2024. A total of 1207 individuals (580 in Mahalaxmi and 627 in Godawari) were involved. Data on demographics, knowledge, attitude, practice, and adherence was collected using a pre-validated structured questionnaire.

### Results

The average age, educational status, monthly income, occupation, presence of respiratory disease, chronic diseases and communicable diseases were found to be not different among the two municipalities. Work experience, presence of respiratory disease and of health worker in the household was different in the baseline survey among the two locations. Knowledge was higher in Mahalaxmi municipality, but adherence was higher in Godawari municipality (p <0.0001). No significant difference was seen in attitude and practice scales. Knowledge, attitude, practice and adherence scores among different subgroups of respondents in the two municipalities were found to be significantly different for occupation

**Data Availability Statement:** The data is deposited as an Excel file in Figshare. The link is Shankar, P

Ravi; Jha, Nisha; Kafle, Sajala; Joshi, Mili; Pandey, Aakriti; Koirala, Prakriti; et al. (2024). Baseline data on KAP and adherence to antimicrobial therapy in two municipalities of Lalitpur district, Nepal. figshare. Dataset. https://doi.org/10.6084/m9. figshare.27610713.v1.

**Funding:** University Grants Commission collaborative research grant. Grant number CRG-79/80-HS-01. The funders had no role in the study design, data collection and analysis, decision to publish, or preparation of the manuscript.

**Competing interests:** All authors have no conflict of interest except one author, Pathiyil Ravi Shankar, who is an academic editor at PLoS One.

(p <0.0001), and education (p <0.0001). The attitude scores were also significantly different according to presence/absence of respiratory disease in the household (p = 0.027).

## Conclusion

At baseline the two study sites were broadly comparable in terms of participants' demographic characteristics. There was higher knowledge and lower adherence in Mahalaxmi municipality. An educational intervention to improve KAP and adherence is required and will be conducted.

## Introduction

Irrational use of antimicrobials as well as antimicrobial resistance (AMR) are important global issues [1]. The use of antimicrobials has steadily increased globally [2].

According to Murray et al., bacterial antibiotic resistance was linked to 4·95 million fatalities in 2019 [3]. One of the areas with the highest rates of AMR is South Asia. The majority of the worldwide disease burdens linked to and caused by AMR were caused by lower respiratory and thoracic infections, bloodstream infections, and intra-abdominal infections. AMR may cause up to 3.8% of the world's gross domestic product to be lost by 2050, according to World Bank estimates [4]. An estimate of AMR's cumulative effects on food systems and human health could push up to 24 million people into extreme poverty by 2030 [5]. Eight South Asian and Southeast Asian nations were the subjects of an AMR surveillance study [6]. AMR data from low-income nations have a number of issues, including underutilisation of microbiology services, a lack of representativeness, and a lack of standardised laboratory procedures.

Antimicrobials are frequently imported into Nepal as supplements or additions for animal feed. High frequencies of methicillin-resistant Staph aureus (MRSA) (>45% in tertiary care settings), rising carbapenem resistance in gram-negative bacteria (>30% in certain facilities), and developing resistance in Salmonella and E. coli strains were all noted by the National Public Health Laboratory in 2022 [7]. In Nepal, there were 23,200 AMR-related deaths and 6,400 AMR-attributable deaths in 2019. Klebsiella pneumoniae, Staphylococcus aureus, Escherichia coli, Pseudomonas aeruginosa, and Streptococcus pneumoniae were the five most prevalent infections linked to antimicrobial resistance (AMR) [8].

Antimicrobials are readily available without a prescription in poor nations [9–12]. Infectious diseases are more prevalent. Therefore, the issue of irrational usage and cheap access to antibiotics have to be taken very seriously [11–14]. AMR may result in treatment failure, a rise in mortality, and an increase in medical expenses [15].

Involving the community in understanding the issue of AMR and its prevention strategies can be achieved through community involvement. A participatory method for finding, creating, and putting into practice community-led, long-lasting solutions to issues that affect them is community engagement. Involving the community can be a crucial strategy for lowering the issue of AMR [16]. In 2016, Nepal's national antimicrobial resistance containment action plan was created [17].

The Nepalese government has already taken action to control AMR and encourage the prudent use of antibiotics. To address the issue of AMR and lessen its effects, including illness, mortality, and economic impact, the National Action Plan for Antimicrobial Resistance (NAP-AMR) was developed [18]. Antibiotic resistance is also exacerbated by the easy access to antibiotics as over-the-counter medications [19]. Antibiotics are readily available without a

prescription in poor nations [10, 12, 20]. Antimicrobial resistance can lead to treatment failure, increased mortality, and higher healthcare expenditures [21, 22].

Mahalaxmi and Godawari are two neighboring municipalities in Lalitpur district of Nepal. Mahalaxmi Municipality occupies an area of 26.5 square km. The municipality has around 62 thousand population. The Godawari municipality is larger occupying an area of 96.11 sq.km with total population being around 116 thousand. A study found that around 82% of respondents in Mahalaxmi seek modern health care for their illness. Government hospitals, private hospitals and clinics were the preferred sites. Around 12% of respondents self-medicated for their illness [23].

The details of the disease burden in the two municipalities and sources of healthcare are shown in the S1 Table.

Female community health volunteers (FCHVs) are female volunteers involved in advocating healthy behaviors of mothers and other people in the community for promoting safe motherhood, promoting child health and family planning related issues. They promote community-based health promotion and deliver healthcare services. FCHVs and associated mother's groups have an important role in communicating healthcare related interventions to the households [24]. Urban areas are riskier places for AMR than rural areas due to factors like the dense population, and easier access to the community pharmacy outlets for medicines including antibiotics. A mother's group is a group of mothers under the FCHVs representing the households of a community.

Following the baseline data collection, FCHVs and mother's groups will be used to spread awareness of AMR among the community in two urban municipalities of Lalitpur district, Nepal.

Interventions have been used to improve KAP and the use of antibiotics. In Nepal, a community intervention utilizing journalists, schoolteachers, and school children resulted in a significant increase in knowledge of antibiotics [25]. In Hong Kong, two weekly sessions and a peer network using Facebook resulted in better knowledge about the indications of antibiotics and the importance of continuing the full course of treatment at the original dose [26]. At a family health center in the United Arab Emirates an educational intervention and a booklet significantly improved mothers' awareness, practice and attitudes regarding antibiotic use [27]. We will use educational interventions in the next phase of the study.

Mothers are vital members of the family, and their education will help them better comprehend issues and share their knowledge with their families, communities, society, and country. There has never been any prior research done in Nepal on KAP, antibiotic adherence, and AMR among female populations. This is significant since FCHVs are vital components of the healthcare system, and each one is associated with a mother's group. Research has indicated that women may engage in self-medication at a higher rate than men [28]. According to a study, women are tasked with taking care of the family's health and may use antibiotics to treat themselves and their kids [29].

The present study is being done in a community as this may be the right place to educate and communicate about the responsible use of antimicrobials and the burden due to AMR with an aim to assess the knowledge, attitude, practice, and adherence toward antimicrobials among women living in two urban municipalities in Lalitpur district, Nepal.

## Methods

### Research design

A population survey was conducted in two urban municipalities of Lalitpur district, Mahalaxmi (proposed intervention) and Godawari (proposed comparison site). Lalitpur was

selected for this study mainly due to convenience as the researchers were working in Lalitpur district and this made it easier to do the study there. Also, Lalitpur district is part of the Kathmandu valley and has people from all over the country.

## Study duration

The duration of the baseline study was from September 2023 to January 2024.

## Sampling technique

All women belonging to the mother's group of selected FCHVs of the two municipalities were included in the baseline survey.

The number of females in the Mahalaxmi municipality was 60,717 and that in Godawari municipality was 49493. The age group of 40–49 years of age group was 13538 in Godawari municipality, with 15425 being illiterate. Similarly, for the Mahalaxmi municipality, the age group of 40–49 years was 17041 and 11344 were illiterate [30, 31]. All FCHVs from Mahalaxmi municipality were selected. In Godawari, 45 of the 114 FCHVs were randomly selected. The number of mothers associated with a FCHV varied from 10 to 25.

## Sample size

The sample size was 1207 mothers, 580 mothers in Mahalaxmi and 627 mothers in Godawari municipalities. This was calculated based on the prevalence of adherence to antibiotics of 39.1% in an Ethiopian study [32].

## Study site

The study was carried out in the Lalitpur district, of Kathmandu valley in the Bagmati province. The first political division in Nepal is a province, followed by districts and each district is divided into rural or urban municipalities followed by wards. Information about the two municipalities has been provided in the Introduction.

## Ethical approval and informed consent

Ethical approval for the study was obtained from the Ethical review board of Nepal Health Research Council on 29[th] August 2023 including approval for the baseline study with a reference number 296 and protocol registration number 535/2023. Written informed consent was obtained from each participant prior to the start of the study. All the ethical principles laid down by the approving body were followed strictly.

## Data collection tool

The questionnaire was developed based on previously published studies [33–35]. A few modifications were made in the questionnaire according to the study objectives.

## Questionnaire validation

The English version of the questionnaire was discussed in the study team to establish the validity of the contents. This questionnaire was then forward translated to Nepali by a bilingual expert (fluent in both Nepali and English) and back translated to English by another bilingual expert. The two versions (the original and the back translated version) were then compared. The original content validated questionnaire, and the back translated questionnaires were discussed among study team members and the final Nepali version of the study tool was finalized.

The face validity of the questionnaire was established by obtaining opinions of five mothers who were not involved in the study. The questionnaire had five sections where the first section was for demographic characteristics of the respondents. The second section was for the knowledge on antimicrobials and antimicrobial resistance, third for attitude towards antimicrobial resistance, and fourth for practice for using antimicrobials. The fifth section was for the assessment of medication adherence for the antibiotic therapy. There were 18 questions for the knowledge section, 12 questions for the attitude section and 5 questions for the practice section. There were 10 questions for the assessment for antibiotic adherence.

## Data collection, management and analysis

The data was collected using KOBO Toolbox using android tablets by the research team and trained data collectors. The collected data was imported, cleaned and analyzed using SPSS software version 29 [36].

The demographic and personal characteristics were analyzed descriptively using measures of central tendency, variation and frequencies as appropriate. The AMR knowledge was compared among respondents from the two municipalities. For the attitude scores the degree of agreement with statements was compared. The total knowledge score was calculated by providing a score of 1 for a correct answer and 0 for an incorrect one. For the attitude statements a score of 5 was given for strongly agree, 4 for agree, 3 for neutral, 2 for disagree and 1 for strongly disagree. The scoring was reversed for negatively worded statements. For the practice statements the frequency of never, rarely, sometimes, often and always were scored as 1, 2, 3, 4, 5 or vice versa depending on the statements. For the adherence section the score was measured using yes or no responses to statements. The total knowledge, attitude, practice and adherence scores were calculated. The normality of distribution of these scores was calculated using one-sample Kolmogorov Smirnov test ($p < 0.05$). Appropriate measures of central tendency and variation were calculated, and these compared among different subgroups of respondents in the two municipalities.

## Results

There were 580 women from Mahalaxmi and 627 women from Godawari municipality who completed the questionnaire. The average age, educational status, monthly income, occupation, presence of chronic diseases and communicable diseases were found to be not different among the two municipalities. Work experience, presence of respiratory disease and of health worker in the household was different in the baseline survey among the two locations as shown in Table 1 below. The median age of the participants in Mahalaxmi municipality was 45 years and that of Godawari municipality was 42 years. Maximum participants, 327 (52.2%) from Godawari municipality were having no formal education. Many participants, 327 (52.2%) from Mahalaxmi municipality were not working and 346 (59.7%) participants were homemakers, whereas 95 (15.2%) participants were not working and 416 (66.3%) were homemakers from Godawari municipality. Many participants, 114 (19.7%) were having no income from Mahalaxmi municipality and 135 (21.5%) were having no income from Godawari municipality.

Table 2 shows the response to the knowledge questions that were significantly different among the participants from the two municipalities. There was a significant difference in the understanding of amoxycillin as an antibiotic between the two municipalities. Similarly, the participants were aware that using antibiotics can lead to secondary infections by killing good bacteria in our bodies. They had heard the term antibiotics [529 (84.4%) from Godawari and

**Table 1. Socio demographic variables by municipalities, AMR baseline survey 2024.** [n = 1207].

| Variable | Median ± IQR | p-value |
|---|---|---|
| Age (in years) | | |
| Mahalaxmi (n = 580) (n, %) | 45 ± 17 | 0.800 |
| Godawari (n = 627) (n, %) | 42 ± 14 | |
| Work experience in years | | |
| Mahalaxmi | 10 ± 17 | 0.526 |
| Godawari | 10 ± 14 | |
| Number of household members | | |
| Mahalaxmi | 4 ± 2 | 0.491 |
| Godawari | 4 ± 1 | |
| Number of household members with chronic disease | | |
| Mahalaxmi | 1 ± 0 | 0.762 |
| Godawari | 1 ± 1 | |
| **Variable** | **Yes (n, %)** | **P value** |
| Presence of respiratory disease in the household | | |
| Mahalaxmi | 479 (83.0%) | **0.044** |
| Godawari | 487 (78.4%) | |
| Presence of other communicable diseases in the household | | |
| Mahalaxmi | 565 (97.4%) | 0.970 |
| Godawari | 611 (97.4%) | |
| Presence of chronic illness (hypertension, diabetes) in the household | | |
| Mahalaxmi | 304 (52.5%) | 0.341 |
| Godawari | 311 (49.8%) | |
| No disease present in the household | | |
| Mahalaxmi | 388 (66.9%) | **<0.001** |
| Godawari | 477 (67.2%) | |
| Presence of health worker at household | | |
| Mahalaxmi | 492 (84.8%) | **0.032** |
| Godawari | 558 (89.0%) | |

The values in bold font indicate there are significant differences in values between the groups.

506 (87.2%) from Mahalaxmi]. They also knew that paracetamol and the drugs used to treat gastric conditions are not antibiotics.

Respondents were unable to differentiate antibiotics from other medicines. Mothers were unaware that taking antibiotic as self-medication can be one of the reasons for AMR, and that antimicrobials are the medicines that are used to kill or inhibit growth of bacteria. They were also not aware about AMR being an important and serious global public health problem.

Mothers from the two municipalities showed a significant difference in their agreement with the seriousness of antimicrobial resistance as a serious issue globally. The attitude was also different for the statement on the responsibility of the pharmacists to educate the patient on proper use of antimicrobials and antibiotic can be dispensed without prescription.

Mothers had a good score for the statement that the new antibiotic development can solve antimicrobial resistance issue. They were also convinced that the patients should visit a physician before starting any antibiotic and need not see the physicians for minor infections. Table 3 shows the attitude scores that were significantly different among the mothers of the two municipalities.

**Table 2. Responses to knowledge statements significantly different among respondents from the two municipalities.**

| Variable | Yes (n, %) | p-value |
|---|---|---|
| Can antibiotics cause secondary infections after killing good bacteria present in our bodies? | | |
| Mahalaxmi | 212 (36.6%) | <**0.001** |
| Godawari | 169 (27.0%) | |
| I can recognize antibiotics in my prescription as I always differentiate antibiotics from other medicines. | | |
| Mahalaxmi | 217 (37.4%) | <**0.001** |
| Godawari | 143 (22.8%) | |
| Household storage of antibiotics for future illness can develop antibiotic resistance. | | |
| Mahalaxmi | 230 (39.8%) | <**0.001** |
| Godawari | 181 (29.0%) | |
| Antimicrobials are any medicament used to kill or inhibit growth of bacteria. | | |
| Mahalaxmi | 198 (34.3%) | **0.012** |
| Godawari | 172 (27.6%) | |
| If antimicrobials are taken frequently, it may stop working in the future. | | |
| Mahalaxmi | 315 (54.5%) | <**0.001** |
| Godawari | 233 (37.3%) | |
| Antibiotic resistance is an important and serious public health problem in the world. | | |
| Mahalaxmi | 222 (38.5%) | <**0.001** |
| Godawari | 172 (27.7%) | |
| Common cold can be treated with antibiotics | | |
| Mahalaxmi | 201 (34.8%) | <**0.001** |
| Godawari | 163 (26.1%) | |
| Antibiotics are used to reduce pain. | | |
| Mahalaxmi | 274 (47.4%) | **0.005** |
| Godawari | 245 (39.3%) | |
| Antibiotics can cause side effects (allergies, diarrhea, vomiting). | | |
| Mahalaxmi | 340 (58.9%) | <**0.001** |
| Godawari | 302 (48.3%) | |
| Antibiotic resistance is the loss of sensitivity of antibiotics to a specific bacterium. | | |
| Mahalaxmi | 192 (33.4%) | <**0.001** |
| Godawari | 134 (21.4%) | |

Table 4 shows the frequency of carrying out different actions related to antibiotics that were significantly different among women from the two municipalities.

Table 5 shows the answers to statements on adherence to antibiotics that were significantly different among respondents in the two municipalities.

Table 6 shows the knowledge and attitude scores for different subgroups of respondents. The results showed that the scores were found to be significantly different with regard to education (p <0.0001), and occupation (p <0.0001).

Similarly, the attitude scores for the groups according to presence/absence of respiratory disease in the household were also found to be significantly different (p = 0.027). Table 7 shows the practice and adherence scores that were significantly different among the respondents.

## Discussion

Antimicrobial resistance (AMR) is a worldwide public health problem particularly in low- and middle-income countries (LMICs) [37]. This study evaluated respondents' knowledge of

**Table 3. Attitude scores that were significantly different among respondents from the two municipalities.**

| Variable | Strongly agree | Agree | Neutral | Disagree | Strongly disagree | P value |
|---|---|---|---|---|---|---|
| | | | Number (percentage) | | | |
| Antimicrobial resistance has become a serious issue all over the globe | | | | | | |
| Mahalaxmi | 16 (2.8%) | 254 (43.9%) | 273 (47.2%) | 34 (5.9%) | 1 (0.2%) | **<0.001** |
| Godawari | 1 (0.2%) | 197 (31.4%) | 274 (43.7%) | | 2 (0.3%) | |
| It is the responsibility of pharmacists to educate the patient on proper use of antimicrobials | | | | | | |
| Mahalaxmi | 19 (3.3%) | 310 (53.7%) | 175 (30.3%) | 71 (12.3%) | 2 (0.3%) | **<0.001** |
| Godawari | 18 (2.9%) | 264 (42.2%) | 170 (27.2%) | 173 (27.6%) | 1 (0.1%) | |
| New antibiotic development can solve antimicrobial resistance issue | | | | | | |
| Mahalaxmi | 6 (1.0%) | 140 (24.3%) | 356 (61.9%) | 72 (12.5%) | 1 (0.2%) | **<0.001** |
| Godawari | 2 (0.3%) | 108 (17.3%) | 364 (58.2%) | 148 (23.7%) | 3 (0.5%) | |
| Antibiotic can be dispensed without prescription | | | | | | |
| Mahalaxmi | 9 (1.6%) | 104 (18.0%) | 89 (15.4%) | 272 (47.1%) | 104 (18.0%) | **0.002** |
| Godawari | 4 (0.6%) | 67 (10.7%) | 92 (14.7%) | 339 (54.3%) | 122 (19.6%) | |
| Patients should be requested to consult a physician before dispensing an antibiotic without prescription | | | | | | |
| Mahalaxmi | 108 (18.7%) | 332 (57.4%) | 85 (14.7%) | 53 (9.2%) | 0 (0.0%) | **<0.001** |
| Godawari | 38 (6.1%) | 388 (61.9%) | 94 (15.0%) | 106 (16.9%) | 1 (0.2%) | |
| Patients with minor infections need not consult a physician for an antibiotic | | | | | | |
| Mahalaxmi | 14 (2.4%) | 304 (52.7%) | 114 (19.8%) | 140 (24.3%) | 5 (0.9%) | **0.008** |
| Godawari | 3 (0.5%) | 333 (53.5%) | 102 (16.4%) | 182 (29.2%) | 3 (0.5%) | |
| Patients with minor infections can be dispensed antibiotics without prescription by pharmacists | | | | | | |
| Mahalaxmi | 11 (1.9%) | 315 (54.5%) | 130 (22.5%) | 120 (20.8%) | 2 (0.3%) | **0.004** |
| Godawari | 3 (0.5%) | 354 (56.7%) | 103 (16.5%) | 161 (25.8%) | 3 (0.5%) | |
| Tackling antibiotic resistance is solely the responsibility of physicians | | | | | | |
| Mahalaxmi | 11 (1.9%) | 202 (35.0%) | 256 (44.4%) | 103 (17.9%) | 5 (0.9%) | **0.007** |
| Godawari | 1 (0.2%) | 217 (34.3%) | 257 (41.1%) | 142 (22.7%) | 9 (1.4%) | |
| One of the reasons for dispensing antibiotics without prescription maybe the business benefit of pharmacy | | | | | | |
| Mahalaxmi | 33 (5.7%) | 327 (56.8%) | 140 (24.3%) | 75 (13.0%) | 1 (0.2%) | **<0.001** |
| Godawari | 19 (3.0%) | 338 (54.0%) | 120 (19.2%) | 147 (23.5%) | 2 (0.3%) | |
| Reasons for dispensing antibiotics without prescription maybe no time and budget | | | | | | |
| Mahalaxmi | 12 (2.1%) | 331 (57.4%) | 155 (26.9%) | 79 (13.7%) | 0 (0.0%) | **<0.001** |
| Godawari | 16 (2.6%) | 333 (53.2%) | 124 (19.8%) | 152 (24.3%) | 1 (0.2%) | |
| Reasons for dispensing antibiotics without prescription maybe the competency of community pharmacists to treat common infections. | | | | | | |
| Mahalaxmi | 7 (1.2%) | 192 (33.3%) | 289 (50.2%) | 88 (15.3%) | 0 (0.0%) | **<0.001** |
| Godawari | 12 (1.9%) | 255 (40.9%) | 195 (31.3%) | 160 (25.7%) | 1 (0.2%) | |
| Reasons for dispensing antibiotics without prescription maybe the patient's requests for antibiotics | | | | | | |
| Mahalaxmi | 11 (1.9%) | 250 (43.3%) | 224 (38.8%) | 91 (15.8%) | 1 (0.2%) | **<0.001** |
| Godawari | 6 (1.0%) | 300 (48.1%) | 167 (26.8%) | 149 (23.9%) | 2 (0.3%) | |

antimicrobials and antibiotic resistance as well as attitudes, the practices and the adherence of antimicrobial use.

## Knowledge of respondents

There was a significant difference in the understanding of amoxycillin as an antibiotic between the respondents from the two municipalities. The participants were aware that the use of antibiotics can lead to the development of secondary infections. This finding was better than the findings from another study, where the participants' knowledge of the medicine was based on

**Table 4. Frequency of carrying out different actions related to antibiotics were significantly different among respondents in the two municipalities.**

| Variable | Never | Rarely | Sometimes | Often | Always | P value |
|---|---|---|---|---|---|---|
| Have you ever educated someone on when and how to use the antibiotic? | | | | | | |
| Mahalaxmi | 364 (63.0%) | 14 (2.4%) | 146 (25.3%) | 39 (6.7%) | 15 (2.6%) | <0.001 |
| Godawari | 358 (57.1%) | 5 (0.8%) | 120 (19.1%) | 53 (8.5%) | 91 (14.5%) | |
| Have you educated the patient on minor side effects of antibiotics? | | | | | | |
| Mahalaxmi | 462 (80.1%) | 10 (1.7%) | 93 (16.1%) | 10 (1.7%) | 2 (0.3%) | 0.047 |
| Godawari | 535 (85.5%) | 8 (1.3%) | 79 (12.6%) | 4 (0.6%) | 0 (0.0%) | |
| Have you ever used antibiotics to treat minor ailments in patients without a prescription? | | | | | | |
| Mahalaxmi | 500 (86.5%) | 9 (1.6%) | 62 (10.7%) | 6 (1.0%) | 1 (0.2%) | <0.001 |
| Godawari | 581 (92.7%) | 18 (2.9%) | 23 (3.7%) | 5 (0.8%) | 0 (0.0%) | |

the appearance of the antibiotic Amoxicillin was administered as a separate red and yellow colored capsule and was named 'red & yellow medicine' [38].

In Nepal, several factors can play a role in the overuse of antibiotics such as, limited diagnostic facilities and the easy availability of antibiotics as an over-the-counter medicine [39]. Lack of knowledge of antibiotics and problems associated with their irrational use could also contribute. In a study in Mangalore, India, 74.3% of participants did not complete the full course of antibiotics, and 51.82% used leftover antibiotics for the same case [40]. In another study participants agreed that antibiotics are not anti-inflammatory medicines. A percentage of participants considered paracetamol and ibuprofen as an antibiotic in a study done in Saudi Arabia [41].

Antibiotics can reduce pain and inflammation in infections by killing/reducing the causative microorganisms and this may confuse the lay respondents.

Mothers from both the areas were unaware that self-medication with antibiotics can be a reason for AMR, and that the antimicrobials are any medicaments used to kill or inhibit the growth of bacteria. Self-medication can be one of the important factors for development of AMR documented in many studies [41, 42].

**Table 5. Answers to statements on adherence to antibiotics that were significantly different among respondents in the two municipalities.**

| Variable | Yes (n, %) | p-value |
|---|---|---|
| Do you ever forget to take your antibiotics? | | |
| Mahalaxmi | 203 (35.1%) | 0.035 |
| Godawari | 376 (64.9%) | |
| Sometimes if you feel worse when you take antibiotics, do you stop taking it | | |
| Mahalaxmi | 189 (32.7%) | <0.001 |
| Godawari | 142 (22.7%) | |
| I take my antibiotics only when I am sick. | | |
| Mahalaxmi | 115 (19.9%) | <0.001 |
| Godawari | 186 (29.9%) | |
| Overall result | | |
| Adherent (60% or greater of the total score) | 265 (43.6%) | <0.001 |
| Nonadherent (less than 60% of the adherence score.) | 267 (46.3%) | |

The answer for the statement 'Sometimes if you feel worse when you take the antibiotics, do you stop taking it?' showed a significant difference among the mothers of the two municipalities.

The result also shows the knowledge and adherence scores to be significantly different.

**Table 6. Knowledge and attitude scores that were significantly different among different subgroups of respondents (in the two municipalities surveyed).**

| Characteristic | Knowledge score Mean (±SD) | P value | Attitude score Median (IQR) | P value |
|---|---|---|---|---|
| Highest education | | | | |
| No formal education | 8.80 ± 2.6 | <0.001 | 38 (7) | 0.003 |
| Primary | 9.88 ± 2.8 | | 42 (6) | |
| Secondary | 10.33 ± 2.78 | | 41 (8) | |
| Higher secondary | 11.39 ± 2.6 | | 40 (8) | |
| Bachelor's degree and above | 11.62 ± 2.8 | | 40 (8) | |
| Occupation | | | | |
| No work | 9.95 ± 3.03 | <0.001 | 42 (9) | <0.001 |
| Daily wage | 9.98 ± 2.04 | | 39 (7) | |
| Retired | 11.02 ± 2.94 | | 41 (12) | |
| Home maker | 9.40 ± 2.79 | | 40 (7) | |
| Other | 9.47 ± 3.21 | | 40 (7) | |

Limited knowledge about the differentiation of antibiotics and other medicines among the mothers was similar to other study findings [38, 43, 44]. Sharing antibiotics and storing them for future use can also promote self-medication and can add to the development of AMR [45]. Problems in knowledge and practice about antibiotics were noted among 321 breastfeeding mothers in Kaduna state, Nigeria. The problem was also seen among 25.27% of participants from Nigeria who believed that antibiotics can be used to treat fever, common cold and cough, infections caused by viruses and other microorganisms. Many participants, 43.7% had not heard the term antibiotic before and 45.7% did not believe that using antibiotics for mothers can affect the baby. Majority of participants, 74.6% believed that antibiotic resistance is when our body is not responding to the antibiotics which to our study [46].

A study from Mozambique on knowledge and practice of antibiotics noted most participants knew the term 'antibiotic'. Participants related antibiotics with their colors, shapes and sizes and health conditions. They also mentioned that the habits contributing to the development of resistance could possibly be not buying the full course of antibiotics, sharing medicines and not completing the full duration of the treatment. Even in a developed nation like Japan, 80% of the participants did not know that antibiotics do not kill viruses and that antibiotics are ineffective against cold and flu [47].

**Table 7. Practice and adherence scores that were significantly different among different subgroups of respondents (in the two municipalities surveyed).**

| Characteristic | Practice score Median (IQR) | P value | Adherence score Median (IQR) | P value |
|---|---|---|---|---|
| Highest Education | | | | |
| No formal education | 5 (2) | <0.001 | 6 (3) | <0.001 |
| Primary | 5 (4) | | 5 (3) | |
| Secondary | 7 (4) | | 5 (3) | |
| Higher secondary | 9 (6) | | 6 (3) | |
| Bachelor's degree and above | 8 (5) | | 6 (2) | |
| Occupation | | | | |
| No work | 5 (2) | <0.001 | 6 (3) | <0.001 |
| Daily wage | 5 (4) | | 5.5 (2) | |
| Retired | 7 (4) | | 6 (2) | |
| Home maker | 7 (4) | | 5 (3) | |
| Other | 8 (4) | | 5 (3) | |

Many participants had not heard the term 'antibiotic resistance'. They understood the term 'resistance' as a failure of the treatment and the medicine not working for a particular illness [38]. In the Japanese study only about 4 out of 10 participants had heard the term 'AMR' [48]. Similar knowledge gaps about AMR were noted in a study done in Sialkot, Pakistan [49].

Antibiotics are often used in non-bacterial conditions. A study from Bangladesh showed that injectable and oral antibiotics were used in cases of pyrexia (30.5%), loose motions (4.5%), pyrexia with cough and cold (47.6%). Antibiotics like cotrimoxazole (31.0%) and amoxicillin (29.0%) were used for pyrexia with common cough and cold while amoxicillin (11.0%) and cotrimoxazole (14.0%) were taken for pyrexia with diarrhea [50]. Our findings also showed that the mothers were not aware about using antibiotics in diarrhea and pyrexia that may be caused by viruses and other organisms.

## Attitude toward antibiotics

Mothers from both the municipalities showed significant differences on the seriousness of antimicrobial resistance as a global issue. Studies have shown that the mothers had an improper attitude towards using antibiotics [50–52].

In the present study knowledge, attitude, practice and adherence scores among different subgroups of respondents in the two municipalities were found to be significantly different with regard to education (p <0.0001), and occupation (p <0.0001). Problems with attitude and practice toward the use of antibiotics were noted in Turkey among parents of children below 18 years of age with a total of 15.7% of parents stating that antibiotics were used in children with pyrexia [53]. Over one-third of parents mentioned that antibiotics could cure any infections caused due to viruses and 6.3% of parents declared that they pressurize their pediatricians for prescribing some antibiotics; Problems with knowledge and attitude were also noted among the mothers visiting the pediatrics clinics in Riyadh, Saudi Arabia [54]. The authors found only 17.3% of the participants strongly agreed that most upper respiratory tract infections are of viral origin and only 12.3% of the participants correctly identified the use of antibiotics can cause side effects.

## Practice toward antibiotics

Reasons for dispensing antibiotics without prescription may be the business benefit of the pharmacy, lack of time and the patient requests for antibiotics. A study has mentioned that most mothers (89.63%) had poor practice towards antibiotic use [55, 56]. The reason for poor practice may be the poor knowledge of the mothers in the community. In a study in Iran, it was found that mother's knowledge and behavior regarding antibiotic use was intermediate, and behavior was associated with parent's education and mother's occupation [57]. In Peru, parents used to buy antibiotics without prescription to self-medicate their children or following the recommendation of a pharmacist [58]. Antibiotics have been used to treat a variety of menstrual symptoms including cramps, bloating, heavy bleeding, headaches, pimples/acne, moodiness, tender breasts, backache, joint and muscle pain and approximately one in four university women surveyed in southwest Nigeria self-medicate with antibiotics [59]. A systematic scoping review mentions accessibility, affordability, and conditions of health facilities, as well as the health-seeking behavior, as factors that influence self-medication with antibiotics in resource constrained settings [60].

## Adherence and knowledge toward antibiotics

Knowledge was higher in Mahalaxmi municipality, but adherence was higher in Godawari municipality (p <0.0001), but no significant difference was seen in attitude and practice scales.

A study has shown that mothers have higher knowledge levels may have better adherence [61]. Knowledge gaps were noted in Malaysia and most participants did not complete the antibiotic course [62]. Like in our study most respondents incorrectly perceived that antibiotic resistance occurs when the body becomes resistant to antibiotics and AMR is not a problem in Malaysia. In a study in Indonesia over 40% of respondents obtained antibiotics from a pharmacy without a doctor's prescription and over 73% believed that antibiotics could be used to treat viral infections [62, 63].

### Key findings

Limited knowledge about the differentiation of antibiotics and other medicines among the mothers was similar to the findings in other studies. Many participants had not heard the term 'antibiotic resistance'. Knowledge, and adherence scales were higher in Mahalaxmi (p <0.0001), but no significant difference was seen in attitude and practice scales.

### Recommendation

There is an urgent need for educational interventions among the participants for enhancing the knowledge about the safe and rational use of antibiotics. Educational interventions may improve awareness and knowledge, but the challenge might be to retain the information. We will use educational interventions in the next phase of the study.

### Limitations

The major limitation of this study is that it has been conducted only in two municipalities of Lalitpur district. The sampling method is not random hence the result may not be fully generalizable.

### Conclusion

Improving knowledge of the general public can be considered as one of the important approaches for addressing AMR. Educational intervention for the antimicrobial stewardship and the rational use of antibiotics can play a vital role towards reducing AMR. Designing an intervention needs the background knowledge of the target people for a continuous assessment for the effectiveness of the intervention. At baseline the two study sites were broadly comparable in terms of participants' demographic characteristics. There was higher knowledge and lower adherence in Mahalaxmi municipality compared to Godawari municipality. Educational intervention will be done for the FCHVs discussing different case scenarios related to the use of antimicrobial agents.

### Supporting information

**S1 Table. These are the indicators for Godawari and Mahalaxmi.**
(DOCX)

**S2 Table. This is the overall table for Tables 1 to 7.**
(DOCX)

**S1 Questionnaire. Questionnaire for baseline survey.**
(DOCX)

**S1 Checklist. STROBE statement—Checklist of items that should be included in reports of** *cross-sectional studies.*
(DOC)

**S2 Checklist. Inclusivity in global research.**
(DOCX)

## Acknowledgments

The authors would like to thank all the participants from both the municipalities. We would also like to thank the ward chairs of the Godawari and Mahalaxmi municipalities for allowing us to conduct the research in their areas.

## Declaration

The authors had used the software Quillbot for paraphrasing certain parts of the manuscript.

## Author Contributions

**Conceptualization:** Nisha Jha, Sajala Kafle, Mili Joshi, Aakriti Pandey, Prakriti Koirala, Shital Bhandary, Pathiyil Ravi Shankar.

**Data curation:** Nisha Jha, Aakriti Pandey, Prakriti Koirala, Shital Bhandary.

**Formal analysis:** Shital Bhandary.

**Methodology:** Nisha Jha, Sajala Kafle, Mili Joshi, Aakriti Pandey, Prakriti Koirala, Shital Bhandary, Pathiyil Ravi Shankar.

**Project administration:** Nisha Jha, Sajala Kafle, Mili Joshi, Aakriti Pandey, Prakriti Koirala, Shital Bhandary.

**Supervision:** Nisha Jha, Sajala Kafle, Mili Joshi, Aakriti Pandey, Prakriti Koirala, Shital Bhandary.

**Validation:** Nisha Jha.

**Writing – original draft:** Nisha Jha, Pathiyil Ravi Shankar.

**Writing – review & editing:** Pathiyil Ravi Shankar.

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
