## [Decision Letter · Decision Letter 0]

23 Sep 2024

PONE-D-24-31683Knowledge, attitude, practice, and adherence toward antimicrobials among women living in two urban municipalities in Lalitpur district, NepalPLOS ONE

Dear Dr. Jha,

Thank you for submitting your manuscript to PLOS ONE. After careful consideration, we feel that it has merit but does not fully meet PLOS ONE’s publication criteria as it currently stands. Therefore, we invite you to submit a revised version of the manuscript that addresses the points raised during the review process.

Please address the suggestions and comments from reviewers and the editor. 

We look forward to receiving your revised manuscript.

Kind regards,

Kshitij Karki, MPH, MA

Academic Editor

PLOS ONE

“University Grants Commission collaborative research grant. Grant number CRG-79/80-HS-01.”

“All authors have no conflict of interest except one author, Pathiyil Ravi Shankar, who is an academic editor at PLoS One.”

“Authors would like to thank all the participants from both the municipalities. We also like to thank the ward chairs of the Godavari and Mahalaxmi municipalities for allowing us to conduct the research in their areas. We also thank the University Grants Commission for grants us with the collaborative research grant with the grant number CRG-79/80-HS-01.”

“University Grants Commission collaborative research grant. Grant number CRG-79/80-HS-01.”

5. In the online submission form, you indicated that [The data underlying the study will be made available from the corresponding author upon reasonable request.].

Additional Editor Comments:

The intervention study is very timely and needed. Please do following revision and also go through all the comments from reviewers.

- Check the spelling of Godawari Municipality

- Introduction part - no need to write the methodology but add the importance of the study

- Write details of data analysis in methodology part

- Check the table percentage (whether row or column %) as you are comparing two municipalities

- Revisit the table and try to make it more efficient

Thank you so much.

Reviewers' comments:

Reviewer's Responses to Questions

**Comments to the Author**

1. Is the manuscript technically sound, and do the data support the conclusions?

Reviewer #1: Partly

Reviewer #2: Yes

Reviewer #3: Yes

2. Has the statistical analysis been performed appropriately and rigorously? 

Reviewer #1: Yes

Reviewer #2: Yes

Reviewer #3: Yes

3. Have the authors made all data underlying the findings in their manuscript fully available?

Reviewer #1: Yes

Reviewer #2: Yes

Reviewer #3: Yes

4. Is the manuscript presented in an intelligible fashion and written in standard English?

Reviewer #1: No

Reviewer #2: Yes

Reviewer #3: Yes

5. Review Comments to the Author

Reviewer #1: 1. General

Jha et al assessed KAPs on antimicrobial use in Nepal. The manuscript is potentially relevant but would require major revision to make it fit for publication. It has a lot of information but poorly organized.

2. Introduction

a. Authors described the burden and public health importance of AMR but without statistics. Kindly include statistics at global, regional, and national levels to give weight to the problem

b. What did previous studies find about AMR, knowledge, attitude, and use of antibiotics among community members? What were the gaps? Which gap(s) is the current study seeking to address?

c. Merge lines 79-81 with 97-98

3. Methods

a. How were the study areas classified? What differentiates the study areas as intervention and control?

b. How was the data analyzed? It is important to include a brief on data analysis in this section.

c. More respondents were selected in Godawari municipality compared with Mahalaxmi. Why the unequal sample size?

d. Why was Lalitpur district selected for the study and not any other district in the province or country?

e. Background information on the study site is inadequate. Authors should provide additional information on the health profile of the study area including access to healthcare, disease burden etc

4. Results

a. In lines 154-156 authors reported that presence of respiratory disease was found not to be different among the two municipalities. However, in line 156-158, presence of respiratory disease was found to differ among the two locations. Kindly clarify

b. The results in table 1 have not been adequately described. Authors should give a succinct description of the outputs in the table

5. Discussion

a. The discussion section appears to be an extension or repetition of the results. Authors should reorganize, explain the meaning of the results, relate them to studies done elsewhere, and reasons for any differences observed

b. Lines 313-325 are better placed under introduction

6. Conclusion

a. A paragraph on the key findings should precede the recommendations

Reviewer #2: This study describes the results of a survey conducted in two districts of Nepal. The survey collected information about the awareness regarding antibiotics and antimicrobial resistance, attitude towards the use of antibiotics and adherence to treatment with antibiotics among woman in these two districts, using a structured questionnaire.

Below, please find my comments:

1. Page 4 – Please, define the abbreviation FCHV when you first mention it in the main text of the manuscript.

2. How big is the problem of AMR in Nepal? Are there any data about the resistance of specific bacterial pathogens to specific antibiotics? Please describe the situation in the country in this regard.

3. How similar the selected two districts - Mahalaxmi and Godawari - are in terms of urban vs rural areas, population demographics, prevalence of infectious disease, etc. Please, provide some information about this.

4. Page 5, Methods – I would not characterize this study as quasi-experimental. This is a population survey and would recommend referring to it as such.

5. Page 5 – Sampling techniques: You indicate: “All women belonging to the mother’s group of FCHVs of intervention and comparison areas were included in the baseline survey” – in your assessment, how much are these participants representative of the general population of the two districts you studied?

6. Page 24 – In the limitations section: “No intervention for the community pharmacies in this research project is another limitation.” – since this is a population survey, it was not required to have pharmacies participating in the study. I would not consider it as a limitation.

Reviewer #3: Would like to request a clarification in table 1 regarding the variable 'No disease present in the household' followed by yes and no requires some clarity whether yes means that there is disease or there is no disease. Also if a future study is planned as an intervention clarification regarding why respiratory illness was included as a separate illness along with other communicable diseases.

Would also like to request the authors to add something in the title to indicate that this study is the first part of a larger study which to be done as a quasi-experimental study which would give some more clarity to the title as to why this is being done is as an initial step towards the intervention study being done at a later date.

6. PLOS authors have the option to publish the peer review history of their article (what does this mean?). If published, this will include your full peer review and any attached files.

Reviewer #1: No

Reviewer #2: No

Reviewer #3: No

---

## [Author Response · Author response to Decision Letter 0]

27 Nov 2024

Revision letter

Lalitpur, Nepal

Date: 27th November, 2024

To

The Editor-in-Chief

PLOS ONE

Sub: Submission of the revised version of the manuscript for consideration of publication 

Dear Editor-in-Chief,

We are resubmitting the manuscript “PONE-D-24-31683 - Knowledge, attitude, practice, and adherence toward antimicrobials among women living in two urban municipalities in Lalitpur district, Nepal” after revising it and making necessary corrections for consideration of publication in your esteemed journal. We remain grateful to the reviewers for their constructive comments. 

The response to specific comments is as follows: 

General comments

Comment for Journal requirements:

Response: Thank you. The manuscript has been modified to meet the journal criteria.

Comment-2. Thank you for stating the following financial disclosure:

“University Grants Commission collaborative research grant. Grant number CRG-79/80-HS-01.”

Response: The funders had no role in the study design, data collection and analysis, decision to publish, or preparation of the manuscript. 

Comment-3. Thank you for stating the following in the Competing Interests section:

“All authors have no conflict of interest except one author, Pathiyil Ravi Shankar, who is an academic editor at PLoS One.”

Response: We have included the statement ‘This does not alter our adherence to PLOS ONE policies on sharing data and materials’. 

Comment -4. Thank you for stating the following in the Acknowledgments Section of your manuscript:

“Authors would like to thank all the participants from both the municipalities. We also like to thank the ward chairs of the Godavari and Mahalaxmi municipalities for allowing us to conduct the research in their areas. We also thank the University Grants Commission for grants us with the collaborative research grant with the grant number CRG-79/80-HS-01.”

“University Grants Commission collaborative research grant. Grant number CRG-79/80-HS-01.”

Response: The funders information from the acknowledgement section has been deleted as suggested.

Comment -5. In the online submission form, you indicated that [The data underlying the study will be made available from the corresponding author upon reasonable request.].

Response: The data is deposited as an Excel file in Figshare. The link is Shankar, P Ravi; Jha, Nisha; Kafle, Sajala; Joshi, Mili; Pandey, Aakriti; Koirala, Prakriti; et al. (2024). Baseline data on KAP and adherence to antimicrobial therapy in two municipalities of Lalitpur district, Nepal. figshare. Dataset. https://doi.org/10.6084/m9.figshare.27610713.v1

Additional Editor Comments:

The intervention study is very timely and needed. Please do following revision and also go through all the comments from reviewers.

Comment 1: Check the spelling of Godawari Municipality

Response: Thank you. The spelling of Godawari municipality is correct. This is how it has been spelt in English on the government portal http://www.godawarimun.gov.np/en

Comment 2: Introduction part - no need to write the methodology but add the importance of the study

Response: We have mentioned why this study is important. (pages 5 and 9 in the manuscript with track changes)

Comment 3: Write details of data analysis in methodology part

Response: This has been done. (page 15).

Comment 4: Check the table percentage (whether row or column %) as you are comparing two municipalities

Response: We have rearranged the tables. 

Comment 5: Revisit the table and try to make it more efficient

Response: We are only presenting the significantly different values in the tables. This we believe makes it more efficient.

Thank you so much.

Reviewers' comments:

Reviewer's Responses to Questions

Comments to the Author

1. Is the manuscript technically sound, and do the data support the conclusions?

Reviewer #1: Partly

Reviewer #2: Yes

Reviewer #3: Yes

2. Has the statistical analysis been performed appropriately and rigorously?

Reviewer #1: Yes

Reviewer #2: Yes

Reviewer #3: Yes

3. Have the authors made all data underlying the findings in their manuscript fully available?

Reviewer #1: Yes

Reviewer #2: Yes

Reviewer #3: Yes

The data has been deposited on a public repository as mentioned previously. 

4. Is the manuscript presented in an intelligible fashion and written in standard English?

Reviewer #1: No

Reviewer #2: Yes

Reviewer #3: Yes

The manuscript has been copyedited for language and grammar.

5. Review Comments to the Author

Reviewer #1: 1. General

Jha et al assessed KAPs on antimicrobial use in Nepal. The manuscript is potentially relevant but would require major revision to make it fit for publication. It has a lot of information but poorly organized.

We have reorganized the manuscript as suggested by the editor and the reviewers. 

2.Introduction

3.a. Authors described the burden and public health importance of AMR but without statistics. Kindly include statistics at global, regional, and national levels to give weight to the problem

Response: We have addressed this on page 5. 

b. What did previous studies find about AMR, knowledge, attitude, and use of antibiotics among community members? What were the gaps? Which gap(s) is the current study seeking to address?

Response: We have tried to address this on page 9. 

c. Merge lines 79-81 with 97-98

Response: Has been merged as suggested.

3. Methods

a.How were the study areas classified? What differentiates the study areas as intervention and control?

Response: Mahalaxmi and Godawari Municipalities are neighboring municipalities in Lalitpur District of Nepal. The Mahalaxmi Municipality occupies 26.5 square km. The municipality has around 62 thousand population. The Godawari municipality is larger occupying an area of 96.11 sq.km with total population being around 116 thousand. There were no specific reasons for classifying one municipality as control and the other as intervention. 

b.How was the data analyzed? It is important to include a brief on data analysis in this section.

Response: This has been described. (page 15)

c. More respondents were selected in Godawari municipality compared with Mahalaxmi. Why the unequal sample size?

Response: The number of FCHVs in the Godawari municipality was more. The number of mothers associated with mothers’ groups varied from 10 to 25 as mentioned in the appendix. We selected an equal number of FCHVs from both municipalities. There were more mothers associated with the selected mothers’ groups in Godawari as compared to Mahalaxmi. 

Why was Lalitpur district selected for the study and not any other district in the province or country?

Response: Mainly due to convenience as the researchers were working in Lalitpur district and this made it easier to do the study there. Also, Lalitpur district is part of the Kathmandu valley and has people from all over the country. 

e. Background information on the study site is inadequate. Authors should provide additional information on the health profile of the study area including access to healthcare, disease burden etc.

Response: This has been mentioned on page 6 and details of the two study areas are shown in the appendix. 

4.Results

a. In lines 154-156 authors reported that presence of respiratory disease was found not to be different among the two municipalities. However, in line 156-158, presence of respiratory disease was found to differ among the two locations. Kindly clarify.

Response: Thank you for highlighting this mistake. We have corrected this as presence of respiratory disease was found to differ between the two locations. 

c.The results in table 1 have not been adequately described. Authors should give a succinct description of the outputs in the table.

Response: We have added this description in addition to the existing information to provide a succinct description of the outputs in the table. (page 16)

5. Discussion

a. The discussion section appears to be an extension or repetition of the results. Authors should reorganize, explain the meaning of the results, relate them to studies done elsewhere, and reasons for any differences observed

Response: We have reworked certain areas of the discussion. We have mentioned the meaning of the results and compared these to the literature. 

b. Lines 313-325 are better placed under introduction

Response: These lines has been moved as suggested.

5.Conclusion

a.A paragraph on the key findings should precede the recommendations

Response: We have drafted this as suggested (page 45).

Reviewer #2: 

This study describes the results of a survey conducted in two districts of Nepal. The survey collected information about the awareness regarding antibiotics and antimicrobial resistance, attitude towards the use of antibiotics and adherence to treatment with antibiotics among woman in these two districts, using a structured questionnaire.

Below, please find my comments:

Comment- 1: Page 4 – Please, define the abbreviation FCHV when you first mention it in the main text of the manuscript.

Response: We have defined the term FCHV as suggested.

Comment -2: How big is the problem of AMR in Nepal? Are there any data about the resistance of specific bacterial pathogens to specific antibiotics? Please describe the situation in the country in this regard.

We have tried to answer this in the Introduction (page 5). 

Response: 

Comment -3: How similar the selected two districts - Mahalaxmi and Godawari - are in terms of urban vs rural areas, population demographics, prevalence of infectious disease, etc. Please, provide some information about this.

Response: Information about the two municipalities has been added as suggested.

Comment -4: Page 5, Methods – I would not characterize this study as quasi-experimental. This is a population survey and would recommend referring to it as such.

Response: We have corrected the study as a population study. 

Comment -5: Page 5 – Sampling techniques: You indicate: “All women belonging to the mother’s group of FCHVs of intervention and comparison areas were included in the baseline survey” – in your assessment, how much are these participants representative of the general population of the two districts you studied?

Response: The current study is not representative at the general population level as the objective was not to generalize at that level. It is representative of the mothers belonging to the mother's group of these two municipalities. Thus, we can compare socio-demographic and other output and outcome variables among the mothers of these two municipalities. These mothers are not represented at the district level rather they are representative at the selected municipality only.

Comment -6: Page 24 – In the limitations section: “No intervention for the community pharmacies in this research project is another limitation.” – since this is a population survey, it was not required to have pharmacies participating in the study. I would not consider it as a limitation.

Response: This has been corrected as suggested. 

Reviewer #3: 

Comment-1: Would like to request a clarification in table 1 regarding the variable 'No disease present in the household' followed by yes and no requires some clarity whether yes means that there is disease or there is no disease. Also if a future study is planned as an intervention clarification regarding why respiratory illness was included as a separate illness along with other communicable diseases.

Response: No disease present in the household meant about the presence of any disease in the household and yes meant a disease was present in that household while no meant it was not present. Inclusion of respiratory illness as a separate illness was done as the use of antibiotics are commonly seen in these conditions. 

Comment-2: Would also like to request the authors to add something in the title to

---

## [Decision Letter · Decision Letter 1]

22 Dec 2024

Baseline assessment of knowledge, attitude, practice, and adherence toward antimicrobials among women living in two urban municipalities in Lalitpur district, Nepal

PONE-D-24-31683R1

Dear Dr. Nisha Jha,

We’re pleased to inform you that your manuscript has been judged scientifically suitable for publication and will be formally accepted for publication once it meets all outstanding technical requirements.

Kind regards,

Kshitij Karki, MPH, MA

Academic Editor

PLOS ONE

Additional Editor Comments (optional):

Thank you. Please check the the spelling of Godawari Municipality (You have written different (Godavari) in the manuscript).

Reviewers' comments:

Reviewer's Responses to Questions

**Comments to the Author**

1. If the authors have adequately addressed your comments raised in a previous round of review and you feel that this manuscript is now acceptable for publication, you may indicate that here to bypass the “Comments to the Author” section, enter your conflict of interest statement in the “Confidential to Editor” section, and submit your "Accept" recommendation.

Reviewer #2: All comments have been addressed

Reviewer #3: All comments have been addressed

2. Is the manuscript technically sound, and do the data support the conclusions?

Reviewer #2: Yes

Reviewer #3: Yes

3. Has the statistical analysis been performed appropriately and rigorously? 

Reviewer #2: Yes

Reviewer #3: Yes

4. Have the authors made all data underlying the findings in their manuscript fully available?

Reviewer #2: Yes

Reviewer #3: Yes

5. Is the manuscript presented in an intelligible fashion and written in standard English?

Reviewer #2: Yes

Reviewer #3: Yes

6. Review Comments to the Author

Reviewer #2: I have no further comments. Authors have answered all my comments and made corresponding changes in the manuscript. Thank you.

Reviewer #3: (No Response)

7. PLOS authors have the option to publish the peer review history of their article (what does this mean?). If published, this will include your full peer review and any attached files.

Reviewer #2: No

Reviewer #3: No

---

## [Editor Report · Acceptance letter]

26 Dec 2024

PONE-D-24-31683R1 

PLOS ONE

Dear Dr. Jha, 

I'm pleased to inform you that your manuscript has been deemed suitable for publication in PLOS ONE. Congratulations! Your manuscript is now being handed over to our production team.

Kind regards, 

on behalf of

Dr. Kshitij Karki 

Academic Editor

PLOS ONE